# Multi Evaluation of a Modified GoldNano Carb Test for Carbapenemase Detection in Clinical Isolates of Gram-Negative Bacilli

**DOI:** 10.3390/antibiotics11050684

**Published:** 2022-05-18

**Authors:** Arpasiri Srisrattakarn, Aroonlug Lulitanond, Nicha Charoensri, Lumyai Wonglakorn, Suthida Kenprom, Chutipapa Sukkasem, Waewta Kuwatjanakul, Sirikan Piyapatthanakul, Onphailin Luanphairin, Wichuda Phukaw, Kunthida Khanchai, Jantira Pasuram, Chotechana Wilailuckana, Jureerut Daduang, Aroonwadee Chanawong

**Affiliations:** 1Centre for Research and Development of Medical Diagnostic Laboratories, Faculty of Associated Medical Sciences, Khon Kaen University, Khon Kaen 40002, Thailand; arpasr@kkumail.com (A.S.); arolul@kku.ac.th (A.L.); nicha@kku.ac.th (N.C.); wchote@kku.ac.th (C.W.); jurpoo@kku.ac.th (J.D.); 2Clinical Microbiology Unit, Srinagarind Hospital, Khon Kaen University, Khon Kaen 40002, Thailand; wlumya@kku.ac.th (L.W.); suthke@kku.ac.th (S.K.); tirapi@kku.ac.th (C.S.); pwaewt@kku.ac.th (W.K.); 3Clinical Microbiology Laboratory, The Queen Sirikit National Institute of Child Health, Bangkok 10400, Thailand; bloodmackaa@gmail.com; 4Medical Technology Laboratory, Maesot General Hospital, Tak 63110, Thailand; aonphailin.48@gmail.com; 5Clinical Microbiology Laboratory, Lerdsin Hospital, Bangkok 10500, Thailand; wichudapookaw@gmail.com; 6Clinical Microbiology Laboratory, Koh Samui Hospital, Surat Thani 84140, Thailand; k_kunthida@kkumail.com; 7Clinical Microbiology Laboratory, Buriram Hospital, Buri Ram 31000, Thailand; jantira_p@kkumail.com

**Keywords:** *Acinetobacter* spp., AuNP, carbapenem resistance, Enterobacterales, modified GoldNano Carb test, *Pseudomonas aeruginosa*

## Abstract

Carbapenemase-producing Gram-negative bacteria have been increasingly reported. Simple and sensitive methods for carbapenemase detection are still needed. In this study, a gold nanoparticle (AuNP) solution was modified by the addition of zinc sulfate (ZnSO_4_) for improving the conventional GoldNano Carb (cGoldC) test, and the modified GoldC (mGoldC) test was then evaluated for phenotypic detection of carbapenemase production in Gram-negative bacilli clinical isolates. ZnSO_4_ was added to give final concentrations of 0.25, 0.5, 0.75, and 1 mM. The performance of the mGoldC test was evaluated in Enterobacterales, *Acinetobacter* spp., and *Pseudomonas aeruginosa* isolates from six hospitals in different regions using polymerase chain reaction (PCR) as a gold standard. The AuNP solution with 0.25 mM ZnSO_4_ was used for the mGoldC test. Evaluation of the mGoldC test in 495 Enterobacterales, 212 *Acinetobacter* spp., and 125 *P. aeruginosa* isolates (including 444 carbapenemase producers and 388 non-carbapenemase producers) revealed sensitivity, specificity, a positive likelihood ratio, and a negative likelihood ratio of 98.6%, 98.2%, 54.7, and 0.01, respectively. This test is fast, easy to perform, cost-effective (~0.25 USD per test), and highly sensitive and specific for routine carbapenemase detection, thus leading to effective antimicrobial therapy and infection control measures.

## 1. Introduction

Gram-negative bacilli (GNB) such as Enterobacterales, *Pseudomonas aeruginosa,* and *Acinetobacter* spp. with reduced susceptibility to carbapenems by carbapenemase production have been reported worldwide with increasing frequencies [1,2]. Carbapenemases commonly found in these organisms include class A *Klebsiella pneumoniae* carbapenemase (KPC), class B imipenemase (IMP), Verona integron-encoded metallo-β-lactamase (VIM) and New Delhi metallo-β-lactamase (NDM), and class D oxacillinase (OXA) carbapenemases [2,3]. The prevalence rates of carbapenem-resistant GNB isolates vary significantly from one country to another (1–90%) [4,5]. According to National Antimicrobial Resistance Surveillance Center, Thailand (NARST) data, carbapenem resistance rates in *K. pneumoniae* and *Acinetobacter* spp. isolates increased from ~3% and ~45% in 2015 up to ~10 and ~70%, respectively, in 2020 [6]. In addition, the distribution of carbapenemase genes among carbapenem-resistant Enterobacterales (CRE) in Thailand from 2016 to 2018 under the national antimicrobial resistance surveillance system developed by the Thailand National Institute of Health (NIH) was 97% [7]. Thus, detection of carbapenemase production is considered important for clinical practice or infection control purposes.

Currently, various commercial carbapenemase tests that are easy, rapid, highly sensitive, and specific have been introduced into the market [8]. These include colorimetric tests such as the Rapidec Carba NP (bioMerieux Deutschland GmbH, Nürtingen, Germany) and the β-CARBA (BioRad, Marnes-la-Coquette, France) or immunochromatographic tests such as the RESIST-4 O.K.N.V. (Coris BioConcept, Gembloux, Belgium) and the CARBA-5 (CARBA-5, NG biotech, Guipry, France). However, false negatives in isolates with weak or low-level carbapenemase activities were found [8,9,10]. The rapid and accurate MALDI-TOF-MS-based carbapenem hydrolysis assay is also used to detect the carbapenemase activity of bacterial strains [11]. Although the costs of this measurement are low, the equipment remains expensive, thus limiting the wide application of this method in a routine laboratory [12,13]. The price of the rapid commercial kits and the instruments needed for MALDI-TOF-MS remain expensive for low-resource settings.

The Carba NP (CNP) test and modified carbapenem inactivation method (mCIM) have been recommended by the Clinical and Laboratory Standards Institute (CLSI) and are extensively validated worldwide for Enterobacterales and *P. aeruginosa* [14,15,16,17,18]. The mCIM test requires neither special reagents nor equipment but takes 18–24 h. The CNP test is rapid (2 h) and mostly shows high specificity and sensitivity for detecting class A and class B carbapenemases but low sensitivities for OXA carbapenemases [15,17]. In addition, the commercial lysis buffer (B-PERII; Thermo Scientific Pierce, Rockford, IL, USA) used for β-lactamase extraction is costly. The CNP test also failed to detect metallo-β-lactamases (MBLs) (sensitivity of 94% for NDM) [15]. In our previous study, conventional GoldNano Carb (cGoldC) was developed for rapid carbapenemase identification by the use of gold nanoparticles (AuNPs) as a pH indicator for detecting acid production from imipenem hydrolysis [19]. This test is rapid (within 2 h), inexpensive (~0.25 USD per test), and convenient with no requirement of an extraction solution. It provided a sensitivity of 99% for carbapenemase detection in Enterobacterales (100%), *P. aeruginosa* (100%), and *Acinetobacter* spp. (96.7%). However, a false-negative due to very weak carbapenemase activity (OXA) was also observed by the cGoldC test (0.5%). In addition, this method has not yet been evaluated in a large number of clinical isolates of Gram-negative bacilli.

The Zn ion is required for MBL activity and play key roles in the catalytic mechanism [20]. In addition, it can induce a decrease in the inter-nanoparticle distance and electrostatic repulsion invoked by the citrate ligand adsorbed on the surface of the particles [21]. In this study, the cGoldC test was modified by the addition of zinc sulfate (named modified GoldNano Carb or mGoldC test) into the AuNP solution (AuNP-Zn solution) in order to increase its sensitivity for detecting carbapenemase production by accelerating the reaction. The mGoldC test was then evaluated for phenotypic detection of carbapenemase-producing GNB (CPGNB) isolates from various hospitals in different regions of Thailand using PCR as the gold standard.

## 2. Results and Discussion

Among the Enerobacterales isolates from the six hospitals, the NDM rates were higher than those of the OXA-48-like in four hospitals, B, D, E, and F, similar to our previous report in hospital A [22]. In the present study, the majority of OXA-48-like enzymes was observed in hospitals A and C. It was noteworthy that the NDM type was dominant in early surveillance of many regions in Thailand, whereas the OXA-48-like type was likely to increase over the years. Teeraputon et al. also reported that the prevalence rate of OXA-48-like in a hospital from northern Thailand during 2018–2019 was 80% [23]. This trend was also observed in Taiwan, where the OXA-48 rate among carbapenemase- producing *Klebsiella pneumoniae* was increased 6-fold during 2012–2015 [24]. In addition, IMP producers of each *K. pneumoniae* and *Enterobacter* spp. were found in hospital D where NDM was prevalent in 2016 [25]. No KPC enzyme was detected in any hospital. Our results indicate that NDM and OXA-48-like are predominant carbapenemase types, whereas the KPC type has very low prevalence among CRE isolates in Thailand [18,26,27]. For the *P. aeruginosa* isolates, IMP, VIM, and NDM enzymes were found in hospitals A, B, and E, whereas only one VIM producer was reported from hospital D. Interestingly, the NDM type was more prevalent than VIM and IMP in the *P. aeruginosa* isolates from hospital B, suggesting the outbreak of a clonal NDM-producing strain in the hospital. Among the *Acinetobacter* spp. isolates, OXA-23-like, OXA-58-like, and NDM were seen in all four hospitals. This study demonstrated that VIM and IMP were major carbapenemase types in *P. aeruginosa*, whereas OXA-23-like was the most common in *Acinetobacter* spp., similar to previous studies [28,29,30]. In addition, the NDM type was found in various species of Enterobacterales, *Acinetobacter* spp., and *P. aeruginosa* (Appendix A), corresponding to the study of Wu et al. [31]. However, outbreaks of clonal strains may exist in these hospitals, suggesting further investigation. Our limitation due to the small sample size of hospital F should also be noted.

To improve its sensitivity, the cGoldC test was modified by using the AuNP-Zn solution for CPGNB detection. The detection times of the mGoldC using the AuNP-Zn solution with any concentration (0.25–1 mM) were faster than those of the cGoldC (without Zn) (Figure 1). The AuNP solution with either 0.25 or 0.5 mM ZnSO_4_ was still clear with red-wine color (data not shown), but the solution with 0.25 mM ZnSO_4_ had the longest shelf life, at least six months at 4 °C. Thus, the optimal concentration of 0.25 mM ZnSO_4_ was selected for the following experiments. Initial evaluation revealed that the mGoldC detected all carbapenemase producers (29/29) within 1–60 min, whereas 28 from 29 isolates were cGoldC-positive within 1–80 min (Figure 2). One false negative by the cGoldC test was OXA-23-like-producing *A. baumannii*. Therefore, the mGoldC and cGoldC tests provided sensitivity of 100.0% and 96.6%, respectively. The 37 non-CPGNB isolates were negative in both tests, thus giving 100.0% specificity (Table 1). In addition, times to positivity by the mGoldC test were markedly shorter than those by the cGoldC test (Figure 2). Positive results within 5 min by the mGold versus cGoldC were 69.0% versus 41.4% and 63.6% versus 0% for all carbapenemase and OXA producers, respectively (Appendix A). The Zn ion may enhance the exchange of ions between surface negative charges of the citrate-capped AuNPs and the positive charges of acid products from the hydrolysis of imipenem by carbapenemases [21]. It is also an essential cofactor for MBLs such as NDM, IMP, and VIM, thus leading to increased enzyme activity [32,33]. With the use of Zn-supplemented AuNP solution, the sensitivity and speed of the mGoldC method were markedly increased. Girlich et al. also reported higher sensitivity of the modified Hodge test (MHT) for detecting MBL producers by using Zn-containing medium (100 µg/mL ZnSO_4_) [34]. Recently, the immunochromatographic test for NDM showed increased sensitivity when tested using Zn-supplemented Mueller Hinton agar (50 µg/mL ZnSO_4_) [33]. Therefore, the AuNP-Zn solution was used for the detection of CPGNB isolates by the mGoldC test in further evaluation.

The mGoldC test provided overall sensitivity, specificity, positive likelihood ratio (LR+), and negative likelihood ratio (LR-) values of 98.6%, 98.2%, 54.7, and 0.01, respectively (Table 2). Its performance for CPE detection in each hospital showed 94.9–100.0% sensitivity. The mGoldC test failed to detect three NDM-producing *K. pneumoniae* isolates and one OXA-48-like-producing *K. pneumoniae* isolate. Tijet et al. suggested that false negative results of the Carba NP test were associated with strains presenting mucoid colonies [35]. The mGoldC test had excellent sensitivity of 100.0% for the *P. aeruginosa* isolates from the six hospitals. It also provided good performance for detecting carbapenemases in the *Acinetobacter* spp. isolates (96.6–100.0% sensitivity). This test could not detect two *A. baumannii* isolates with either OXA-23-like or OXA-58-like. This is in accordance with the fact that class D carbapenemases have weak carbapenemase activity [16,36,37]. Poirel and Nordmann also reported misidentification of OXA-23-producing *A. baumannii* isolates by the Rapidec Carba NP test [38]. In addition, the false negative by the mGoldC may be due to the small inoculum of 1 µL [19]. Therefore, five loops (1 µL loop) of bacterial colonies were used in this study. Tijet et al. also reported that increasing the bacterial inoculum yielded more positive Carba NP results, particularly in isolates producing OXA carbapenemases [35]. Therefore, the amount of bacteria is critical for phenotypic carbapenemase tests. For the best results of the mGoldC test, we recommend using 5 µL inoculum of bacterial colonies.

Among the six hospitals, the mGoldC test had 90.0–100.0%, 100%, and 83.3–100.0% specificity for all Enterobacterales, *P. aeruginosa,* and *Acinetobacter* spp. isolates, respectively. False positives were found in seven isolates (three *A. baumannii*, three. *E. coli,* and one *K. pneumoniae*) (Table 2) with the detection times of 5–120 min. These may be due to the very weak carbapenemase activity of ESBLs or pAmpCs with or without porin loss, similar to those observed by the Carba NP method [16,39]. Recently, Nordmann et al. reported false positive results by the NitroSpeed-Carba NP test in ACC- or CMY-type AmpC producers [40]. Whitley et al. also observed a false positive of *A. baumannii* with the BD Phoenix CPO Detect test [41]. However, the false positive isolates may contain other known (e.g., SPM, SIM, GIM, GES) or novel carbapenemases. Further investigation by whole-genome sequencing is needed. In addition, the lower specificity for the *Acinetobacter* spp. isolates may be due to the limited numbers of non-carbapenemase producers of these organisms (Table 2). Therefore, evaluation of the mGoldC test in larger samples of *Acinetobacter* spp. is still needed.

## 3. Materials and Methods

### 3.1. Bacterial Collection

A total of 832 non-repetitive GNB clinical isolates, 495 Enterobacterales, 212 *Acinetobacter* spp., and 125 *P. aeruginosa* isolates (444 carbapenemase producers and 388 non-carbapenemase producers), collected from six hospitals in Thailand between February 2019 and August 2020, were included in this study (Appendix A). The six hospitals were as follows: 1670 beds (A) and 900 beds (B) in the Northeast, 500 beds (C) and 435 beds (D) in the central region, 317 beds (E) in the West, and 167 beds (F) in the South.

All isolates were identified by conventional biochemical tests and screened for carbapenemase production by either disc diffusion or minimum inhibitory concentration (MIC) determination methods using criteria as follows: zone diameters of <19 mm in response to either 10 µg imipenem or 10 µg meropenem disks for *P. aeruginosa*, <22 and <18 mm in response to imipenem and meropenem disks, respectively, for *Acinetobacter* spp. according to the CLSI guideline [14], and those of <25 mm in response to meropenem and/or ertapenem disks for Enterobacterales according to the EUCAST (https://eucast.org/clinical_breakpoints/ (accessed on 8 November 2016) guidelines as suggested by Huang et al. [42]. Carbapenem MICs were determined by broth microdilution, and the results were interpreted using the CLSI criteria [14].

This study was approved by the Khon Kaen University Ethics Committee (project number HE561476).

### 3.2. Molecular Detection of Carbapenemase Genes

All isolates were characterized for the presence of carbapenemase genes by the conventional multiplex PCR assays [43,44]. The target genes included *bla*_KPC_, *bla*_NDM_, *bla*_OXA-48-like_, *bla*_VIM_, and *bla*_IMP_ for Enterobacterales; *bla*_NDM_, *bla*_VIM_, and *bla*_IMP_ for *P. aeruginosa*; and *bla*_NDM_, *bla*_OXA-23_, *bla*_OXA-24_, *bla*_OXA-51_, and *bla*_OXA-58_ for *Acinetobacter* spp.

### 3.3. Modification of AuNP Solution

A colloidal suspension of 13 nm AuNP was prepared by using a citrate-reduction method described by Hill and Mirkin [45] with some modifications according to our previous report [19]. The ZnSO_4_ concentrations (0.25, 0.5, 0.75, and 1 mM) in the AuNP solution were optimized, and 0.25 mM ZnSO_4_ was selected for further experiments (AuNP-Zn solution).

### 3.4. Optimization of the mGoldC

The performance of the cGoldC and mGoldC was compared using the strains previously confirmed for their β-lactamases by molecular methods [19]. The bacterial strains included 29 carbapenemase producers (eleven OXA, seven NDM, seven IMP, three VIM, and one KPC-2) and 13 non-carbapenemase producers (four pAmpC, four ESBL & pAmpC, three ESBL, one AmpC, and one non-ESBL and non-AmpC) (Table 1).

The cGoldC test was performed and interpreted as previously described [19] with slight modifications. Briefly, bacterial colonies grown on Mueller–Hinton agar (MHA) (Oxoid, Basingstoke, UK) were scraped by a 1 µL loopful for five times (approximately 5 µL inoculum size), and the reactions were read immediately and at 5, 10, 15, 30, 60, 90, and 120 min. For the mGoldC test, the clinical isolates were tested with the same procedure as the cGoldC test except that the AuNP-Zn solution was used.

### 3.5. Evaluation of the mGoldC Test for Carbapenemase Detection in Six Hospitals

The mGoldC test kits were submitted to the six hospitals. The bacterial colonies of carbapenemase screening-positive isolates from MHA plates were tested following the mGoldC protocol as mentioned above, and the results were recorded. All isolates were submitted to our laboratory for confirmation of carbapenemase genes by medical technologists without any information of the mGoldC results.

The diagnostic parameters of the sensitivity, specificity, an LR+ and LR- of the mGoldC test were calculated by the free software vassarStats (http://vassarstats.net/ (accessed on 11 May 2022) using the PCR method as the gold standard [46].

## 4. Conclusions

The mGoldC test is inexpensive, user-friendly by using bacterial colonies and imipenem/cilastatin powder directly without an extraction buffer, and easy to interpret with a fast and strong positive reaction. The AuNP-Zn solution also has a long shelf-life. Therefore, it could be used as an alternative method for rapid identification of CPGNB.

## Figures and Tables

**Figure 1 antibiotics-11-00684-f001:**
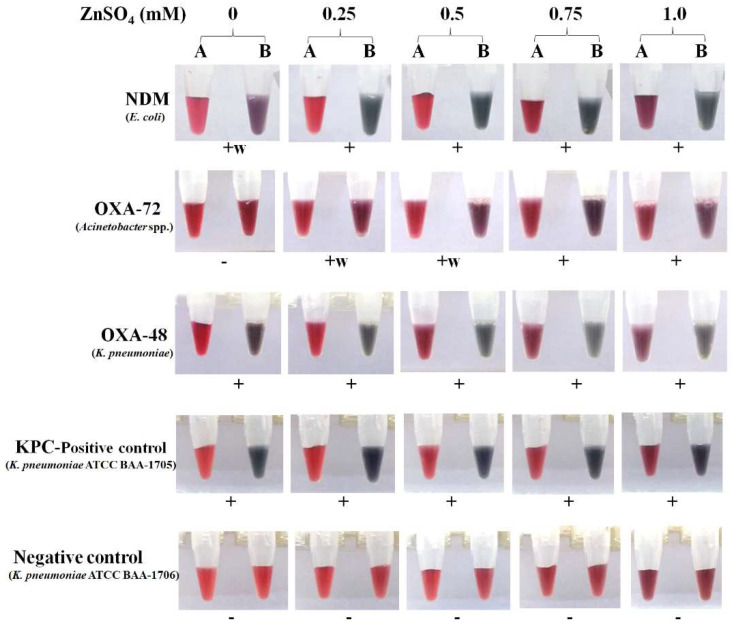
Optimization of ZnSO_4_ (final concentrations of 0, 0.25, 0.5, 0.75, and 1.0 mM) in the AuNP solution for carbapenemase detection by the modified GoldNano Carb test. The reactions were read at 10 min. Tubes A, control (without imipenem); and B, test (with 5 mg/mL imipenem). +, positive result; +w, weakly positive result; -, negative result.

**Figure 2 antibiotics-11-00684-f002:**
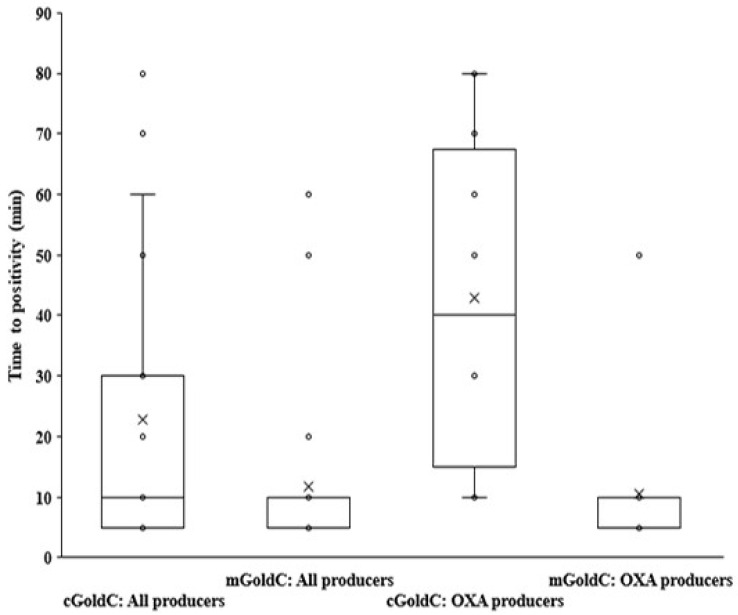
Boxplot of time to positivity by the modified GoldNano Carb test compared with the conventional GoldNano Carb test among 29 carbapenemase producers (all producers) and 11 OXA producers.

**Table 1 antibiotics-11-00684-t001:** Comparison between conventional and modified GoldNano Carb tests for detecting carbapenemase-producing Gram-negative bacilli using PCR as the gold standard.

β-Lactamase Classes and Types (n)	Species (n)	Specified β-Lactamases	No. of Isolates Tested by a			
			cGoldC		mGoldC	
			+ (+w)	-	+ (+w)	-
**Carbapenemase producers (29)**						
**Ambler class A (1)**						
KPC-type (1)	*K. pneumoniae* ATCC BAA-1705 (1)	KPC-2	1		1	
**Ambler class B (17)**						
NDM-type (7)	*K. pneumoniae* (4)	NDM	4		4	
	*E**. coli* (2)	NDM	2		2	
	*A. pittii* (1)	NDM-1	1		1	
IMP-type (7)	*P**. aeruginosa* (2)	IMP-14a	2		2	
	*K**. pneumoniae* (2)	IMP-14a	2		2	
	*P. aeruginosa* (1)	IMP-1	1		1	
	*P. aeruginosa* (1)	IMP-9	1		1	
	*P. aeruginosa* (1)	IMP-15	1		1	
VIM-type (3)	*P. aeruginosa* (3)	VIM-2	3		3	
**Ambler class D (11)**						
OXA-type (11)	*K. pneumoniae* (2)	OXA-48-like	2		2	
	*E. coli* (1)	OXA-48-like	1		1	
	*E. coli* (1)	OXA-181	1		1	
	*A. baumannii* (5)	OXA-23-like	4(3)	1	5(1)	
	*A. baumannii* (1)	OXA-72	1		1	
	*Acinetobacter* spp. (1)	OXA-72	1(1)		1	
**Non-carbapenemase producers (13)**						
ESBL (3)	*K. pneumoniae* (1)	CTX-M-1-like, SHV		1		1
	*E. coli* (1)	CTX-M-1-like, TEM-1		1		1
	*K. pneumoniae* (1) ^b^			1		1
AmpC (1)	*Enterobacter* spp. (1)			1		1
pAmpC (4)	*E. coli* (1)	CMY-2		1		1
	*E. coli* J53 (pSLK54) (1)	ACC-1		1		1
	*E. coli* J53 (pMG251) (1)	ACT-1		1		1
	*K. pneumoniae* (1)			1		1
ESBL & pAmpC (4)	*E. coli* (1)	VEB-like, CMY-8b		1		1
	*E. coli* (1)	VEB-like, MOX-2-like		1		1
	*K. pneumoniae* (1) ^b^			1		1
	*E. coli* (1) ^b^			1		1
Non-ESBL & non-AmpC (1)	*K. pneumoniae* ATCC BAA-1706 (1)			1		1

^a^ cGoldC, conventional GoldNano Carb test; mGoldC, modified GoldNano Carb test; +, positive result; +w, weakly positive result; and -, negative result; ^b^ Positive by double-disc synergy and boronic acid combined disc tests for ESBL and pAmpC producers, respectively.

**Table 2 antibiotics-11-00684-t002:** Diagnostic performance of the modified GoldNano Carb test in Gram-negative bacilli from the six different hospitals.

Hospitals/Organisms	No. Test Isolates	No. of Isolates with PCR		% (95% CI)				No. of Isolates Giving	
		Positive	Negative	Sensitivity	Specificity	LR+	LR−	False Negative (Species, Types)	False Positive (Species)
**A**									
Enterobacterales	105	26	79	100.0 (84.0–100.0)	100.0 (94.2–100.0)	In (NaN-In)	0 (0-NaN)	-	-
*A* *. baumannii*	63	30	33	100.0 (85.9–100.0)	97.0 (82.5–99.8)	33.0 (4.79–227.4)	0 (0-NaN)	-	1 (*A. baumannii*)
*P* *. aeruginosa*	59	9	50	100.0 (62.9–100.0)	100.0 (91.1–100.0)	In (NaN-In)	0 (0-NaN)	-	-
**Total**	227	65	162	100.0 (93.0–100.0)	99.4 (96.1–99.9)	162 (22.95–1143.1)	0 (0-NaN)	-	-
**B**									
Enterobacterales	124	59	65	94.9 (84.9–98.7)	100.0 (93.0–100.0)	In (NaN-In)	0.05 (0.02–0.15)	3 (*K. pneumoniae*, NDM)	-
*A* *. baumannii*	44	38	6	100.0 (88.6–100.0)	83.3 (36.5–99.1)	6 (1.0–35.9)	0 (0-NaN)	-	1 (*A. baumannii*)
*P* *. aeruginosa*	21	14	7	100.0 (73.2–100.0)	100.0 (56.1–100.0)	In (NaN-In)	0 (0-NaN)	-	-
**Total**	189	111	78	97.3 (91.7–99.3)	98.7 (92.1–99.9)	75.9 (10.8–532.1)	0.03 (0.009–0.08)		
**C**									
Enterobacterales	94	71	23	100.0 (93.6–100.0)	95.7 (76.0–99.8)	23 (3.4–156.4)	0 (0-NaN	-	1 (*E. coli*)
**D**									
Enterobacterales	72	20	52	100.0 (79.9–100.0)	100.0 (91.4–100.0)	In (NaN-In	0 (0-NaN)	-	-
*A* *. baumannii*	67	58	9	96.6 (87.0–99.4)	88.9 (50.7–99.4)	8.69 (1.4–55.2)	0.04 (0.01–0.16)	1 (*A. baumannii*, OXA-23); 1 (*A. baumannii*, OXA-58)	1 (*A. baumannii*)
*A. haemolyticus*	1	1	0	100.0 (5.5–100.0)	NaN (NaN-NaN)	NaN (NaN-NaN)	NaN (NaN-NaN)	-	-
*P* *. aeruginosa*	24	1	23	100.0 (5.5–100.0)	100.0 (82.2–100.0)	In (NaN-In	0 (0-NaN)	-	-
**Total**	164	80	84	97.5 (90.4–99.6)	98.8 (92.6–99.9)	81.9 (11.7–574.8)	0.03 (0.006–0.1)		
**E**									
Enterobacterales	53	33	20	100.0 (87.0–100.0)	90.0 (66.9–98.2)	10 (2.7–37.4)	0 (0-NaN)	-	2 (*E. coli*)
*A* *. baumannii*	37	36	1	100.0 (88.0–100.0)	100.0 (5.5–100.0)	In (NaN-In	0 (0-NaN)	-	-
*P* *. aeruginosa*	21	20	1	100.0 (80.0–100.0)	100.0 (5.5–100.0)	In (NaN-In	0 (0-NaN)	-	-
**Total**	111	89	22	100.0 (94.8–100.0)	90.9 (69.4–98.4)	11 (2.9–41.2)	0 (0-NaN)		
**F**									
Enterobacterales	47	28	19	96.4 (79.8–99.8)	94.7 (71.9–99.7)	18.3 (2.7–123.6)	0.04 (0.005–0.3)	1 (*K. pneumoniae*, OXA-48-like)	1 (*K. pneumoniae*)
**Total**									
Enterobacterales	495	237	258	98.3 (95.4–99.5)	98.4 (95.8–99.5)	63.4 (24.0–167.7)	0.02 (0.006–0.05)	4	4
*Acinetobacter* spp.	212	163	49	98.8 (95.2–99.8)	93.9 (82.1–98.4)	16.1 (5.4–48.3)	0.01 (0.003–0.05)	2	3
*P* *. aeruginosa*	125	44	81	100.0 (90.0–100.0)	100.0 (94.4–100.0)	In (NaN-In	0 (0-NaN)	-	-
**Total**	832	444	388	98.6 (96.9–99.4)	98.2 (96.1–99.2)	54.7 (26.2–113.9)	0.01 (0.006–0.03)	6	7

LR+, positive likelihood ratio; LR-, negative likelihood ratio; In, infinity; NaN, the calculation cannot be performed because the values entered include one or more instances of zero. A & B, Northeast; C & D, Central; E, West; F, South.

## Data Availability

Not applicable.

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
