# Peer review of "Multi Evaluation of a Modified GoldNano Carb Test for Carbapenemase Detection in Clinical Isolates of Gram-Negative Bacilli"

_antibiotics, 2022, doi:10.3390/antibiotics11050684_

Round 1

Reviewer 1 Report

2022-04-28

The review of the submission antibiotics-1723552 entitled “Multi evaluation of modified GoldNano Carb test for carbapenemase detection in clinical isolates of Gram-negative bacilli”

Let me state, that the submission is devoted to the very important subject and its conclusions are supported with the acquired  data, but there are some issues with the presentation:

  • Sometimes English is not clear – e.g. “Nowadays, carbapenemase-producing Gram-negative bacteria have been intensifying and are likely to grow over the years” – r. 25. What is that supposed to mean? Or – “as gold standards” – r. 250. I believe that it should be used here in singular as "gold standard” – r.98
  • In the Biology/Medicine journals we should not devote too much attention to the cost-effectiveness – “the cost per test including standard, matrix and imipenem is < 0.10” – r. 70
  • “zinc sulfate (ZnSO4)” – r.93; “zinc-supplemented” – r. 142. I believe that it is not necessary to explain the simplest chemical formulas to the readers, as well as the text will look better if authors replace zinc- for Zn- EVERYWHERE
  • “Enterobacterales” – r.40. According to Wiki: Enterobacterales refers to an order whose nomenclatural type is the genus Enterobacter. https://en.wikipedia.org/wiki/Enterobacterales. Isn’t there any logical reason to do it in non-Latin way only for this taxonomic group?
  • “Rapidec Carba NP (bioMerieux, Nürtingen, Germany)” – r. 62. According to MDPI rules the manufacturer should be mentioned with property type (like LLC or Corp.)

Conclusion – resubmit after correction

Author Response

Thank you very much for your comments, which help improve our manuscript and become more attractive. We have revised our manuscript and responded all comments as in the following.

Comments and Suggestions for Authors

The review of the submission antibiotics-1723552 entitled “Multi evaluation of modified GoldNano Carb test for carbapenemase detection in clinical isolates of Gram-negative bacilli”

Let me state, that the submission is devoted to the very important subject and its conclusions are supported with the acquired data, but there are some issues with the presentation:

  1. Sometimes English is not clear – e.g. “Nowadays, carbapenemase-producing Gram-negative bacteria have been intensifying and are likely to grow over the years” – r. 25. What is that supposed to mean? Or – “as gold standards” – r. 250. I believe that it should be used here in singular as "gold standard” – r.98

Response 1: The sentence has been changed to “Nowadays, carbapenemase-producing Gram-negative bacteria have been growing over the years” (lines 25-26). In addition, “as gold standards” has been changed to “gold standard” as suggested (line 261).

  1. In the Biology/Medicine journals we should not devote too much attention to the cost-effectiveness – “the cost per test including standard, matrix and imipenem is < 0.10” – r. 70

Response 2: We have deleted “(the cost per test including standard, matrix and imipenem is < 0.10 USD)” in lines 72-73.

  1. “zinc sulfate (ZnSO4)” – r.93; “zinc-supplemented” – r. 142. I believe that it is not necessary to explain the simplest chemical formulas to the readers, as well as the text will look better if authors replace zinc- for Zn- EVERYWHERE

Response 3: We have deleted the chemical formulas “ZnSO4” in lines 28 & 97 and replaced zinc- by Zn- for all as suggested.

  1. “Enterobacterales” – r.40. According to Wiki: Enterobacterales refers to an order whose nomenclatural type is the genus Enterobacter. https://en.wikipedia.org/wiki/Enterobacterales. Isn’t there any logical reason to do it in non-Latin way only for this taxonomic group?

Response 4: Enteric Gram-negative bacilli, which were used to refer as family Enterobacteriaceae, have now been classified to 7 different families (Enterobacteriaceae, Erwiniaceae, Pectobactericeae, Yersiniaceae, Hafniaceae, Morganellaceae and Budviciaceae). The order Enterobacterales is therefore used instead of “family Enterobacteriaceae” to include all enteric Gram-negative bacilli (McAdam AJ. J Clin Microbiol 2020; 58: e01888-19). In CLSI guidelines, “family Enterobacteriaceae” has now been replaced with “order Enterobacterales” as well. Genus Enterobacter is one of members in order Enterobacterales and family Enterobacteriaceae.

  1. “Rapidec Carba NP (bioMerieux , Nürtingen, Germany)” – r. 62. According to MDPI rules the manufacturer should be mentioned with property type (like LLC or Corp.)

Response 5: The manufacturer of legal organization of the bioMérieux Group has been changed to “Rapidec Carba NP (bioMerieux Deutschland GmbH, Nürtingen, Germany) in line 64 (file:///C:/Users/Admin/Downloads/LegalDocument.pdf).

Reviewer 2 Report

1. Figure 2, a boxplot with point showing the performance of all 29 carbapenemase producers and 11 OXA producers would be more comprehensive.

Author Response

Thank you very much for your comments, which help improve our manuscript and become more attractive. We have revised our manuscript and responded all comments as in the following.

Comments and Suggestions for Authors

  1. Figure 2, a boxplot with point showing the performance of all 29 carbapenemase producers and 11 OXA producers would be more comprehensive.

Response 1:          We have changed Figure 2 by using a boxplot with point showing the performance of all 29 carbapenemase producers and 11 OXA producers as suggested.

Reviewer 3 Report

This manuscript by Arpasiri Srisrattakarn et al. describes the performance of a modified GoldNano Carb Test for the detection of carbapenemase in GNB.

This manuscript deserves modification before it is eventually accepted.

Global: prefer passive voice - italicize et al - numbers below 12 should be capitalized.

Introduction: What did the author consider "expensive"? This threshold is not obvious to every laboratory operator/medical biologist. Also, the number of readings can be time consuming in a laboratory.

Results: consider presenting the likelihood ratio and not the predictive values (which depend on the prevalence, which is particularly high in Thailand).

Results: please calculate the 95% CI for performance and stratify by center. Also, how was the number of samples to be analyzed predetermined? And the number of centers? And furthermore, how were they selected?

Methods: please consider a single CLSI or EUCAST reference.

Methods: please reference the wassarStats software appropriately.

Author Response

Thank you very much for your comments, which help improve our manuscript and become more attractive. We have revised our manuscript and responded all comments as in the following.

Comments and Suggestions for Authors

This manuscript by Arpasiri Srisrattakarn et al. describes the performance of a modified GoldNano Carb Test for the detection of carbapenemase in GNB.

This manuscript deserves modification before it is eventually accepted.

  1. Global: prefer passive voice - italicize et al - numbers below 12 should be capitalized.

Response 1: We have changed some sentences to be passive voice as suggested (lines 27-29, 84-85 and 153). However, the numbers below 12 have not been capitalized because we did provide the details of the number of isolates showing different carbapenemase types or results in the blanket, for example the number of each species giving false negative or false positive results. We used the same style throughout our manuscript. In addition, “et al.” is not italicize because it is the format (style) of the journal (author instructions).

  1. Introduction: What did the author consider "expensive"? This threshold is not obvious to every laboratory operator/medical biologist. Also, the number of readings can be time consuming in a laboratory.

Response 2: We mean that they are expensive for low-income countries. So we have specified this point in lines 74-75 that “The price of the rapid commercial kits and the instrument of MALDI-TOF-MS remain expensive for low-resource settings.”

  1. Results: consider presenting the likelihood ratio and not the predictive values (which depend on the prevalence, which is particularly high in Thailand).

Response 3: The predictive values has been replaced by likelihood ratio as suggested (lines 35-36, 171-173, 258-259 and Table 2).

  1. Results: please calculate the 95% CI for performance and stratify by center. Also, how was the number of samples to be analyzed predetermined? And the number of centers? And furthermore, how were they selected?

Response 4: We calculated the 95% CI for performance and stratify by center as suggested (Table 2). The number of samples were calculated using the formula of Buderer (Acad Emerg Med 1996; 3: 895-900). The sample sizes ranged from approximately 90-150 samples/hospital. Only the sample of hospital F were smaller than the calculated number. We have added this limitation in lines 127-128.  The number of center were selected from the hospitals in our network locating in each region of Thailand.  We did select different sizes of hospitals (1670-, 900-, 500-, 435-, 317- and 167-bed) for evaluation in order to test whether our test kit could be used in hospitals with different facilities.

  1. Methods: please consider a single CLSI or EUCAST reference.

Response 5: In Thailand, CLSI is generally used as reference. However, in this study the EUCAST criteria were used for the Enterobacterales isolates because both NDM and OXA-48-like group are the major carbapenemase types in Thailand. The EUCAST criteria could detect OXA-48-like producers more than the use of CLSI criteria (Huang et al. J Antimicrob Chemother 2014; 69: 445-50.). We therefore used the CLSI reference for P. aeruginosa and Acinetobacter spp. as usual and the EUCAST criteria for the Enterobacterales. We have cited the study of Huang et al. in line 225.

  1. Methods: please reference the vassarStats software appropriately.

Response 6: We have already provided the reference for the vassarStats software (lines 261 and 429-430).

Round 2

Reviewer 3 Report

The manuscript has been revised according to my previous comment and is now, to my point of view, suitable for publication.